# MODUMER: MODULATING TRANSFORMER FOR IMAGE RESTORATION

## ABSTRACT

Image restoration aims to recover clean images from degraded versions. While Transformer-based approaches have achieved significant advancements in this field, they are limited by high complexity and their inability to capture omni-range dependencies, hindering their overall performance. In this work, we develop **Modumer** for effective and efficient image restoration by revisiting the *Transformer* block and *Modulation* design, which processes input through a convolutional block and projection layers, and fuses features via element-wise multiplication. Specifically, within each unit of Modumer, we integrate the cascaded Modulation design with the downsampled Transformer block to build the attention layers, enabling omni-kernel modulation and mapping inputs into high-dimensional feature spaces. Moreover, we introduce a bioinspired parameter-sharing mechanism to attention layers, which not only enhances efficiency but also improves performance. Additionally, a dual-domain feed-forward network strengthens the representational power of the model. Extensive experiments demonstrate that the proposed Modumer achieves state-of-the-art performance on **ten** different datasets for **five** image restoration tasks: image motion deblurring, image deraining, image dehazing, image desnowing, and low-light image enhancement. Furthermore, our model yields promising performance on all-in-one image restoration tasks.

## 1 INTRODUCTION

As a longstanding task, image restoration aims to recover a high-quality image from its degraded counterpart. It has been quite a challenging problem as infinite solutions correspond to a single input. In recent years, convolutional neural networks (CNNs) have produced promising results on this ill-posed problem by learning direct mappings from the degraded input and restored output (Qin et al., 2020; Ruan et al., 2022; Lee et al., 2021; Liu et al., 2018). However, the shortcomings of convolutional operators are obvious. Due to poor receptive field scaling (Cho et al., 2021; Chen et al., 2024), CNNs are unable to capture long-scale dependencies for powerful image representations.

Recently, Transformers have significantly advanced the state-of-the-art performance of low-level tasks (Song et al., 2022; Chen et al., 2023a; Zamir et al., 2022a). Despite having the great power to capture content-aware global perceptive fields, the self-attention (SA) layer features quadratic complexity to the input, limiting their applications in real-world scenarios. Many attempts have been made to enhance the efficiency of this expensive mechanism. SwinIR (Liang et al., 2021), Uformer (Wang et al., 2022), and Stripformer (Tsai et al., 2022) reduce the complexity of Transformer models by confining the SA operation to a fixed spatial range. Restormer (Zamir et al., 2022a) tactfully switches the operation dimension from the spatial domain to channels. Afterward, a few works explore adopting both channel SA and spatial SA in cascading or parallel manners to improve representational ability (Chen et al., 2024; Zhang et al., 2024; Chen et al., 2023c). Nonetheless, these methods impede the inherent potential of SA, originally proposed for superior global feature modeling, leading to a deterioration in restoration performance. Moreover, they mostly operate within a single scale and cannot capture multi-scale receptive fields within a single unit.

Most recently, the *Modulation* mechanism (Ma et al., 2024b), as illustrated in Figure 1 (b), considering context modeling using a large-kernel convolutional block and modulating the projected input via element-wise multiplication, has become popular in high-level vision tasks (Hou et al., 2024; Guo et al., 2023a; Yang et al., 2022). These approaches are computationally efficient and implement-

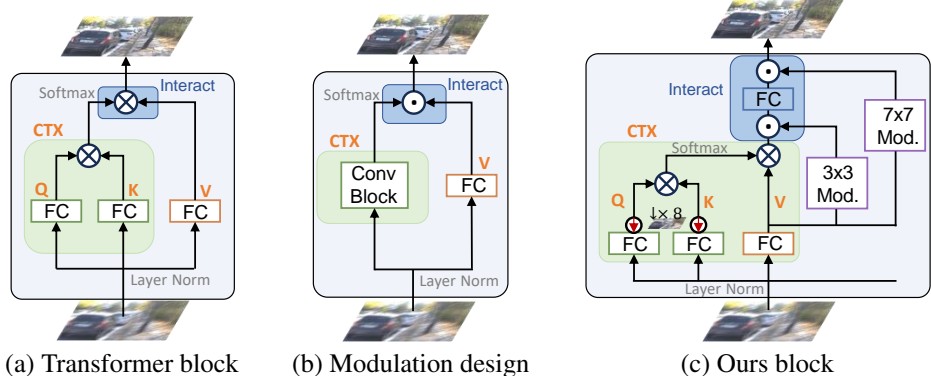

(a) Transformer block     (b) Modulation design     (c) Ours block

Figure 1: Comparison of Transformer block, modulation design, and our attention block. $\otimes$ and $\odot$ are matrix and element-wise multiplication, respectively. Compared to Transformer and modulation blocks, our design performs attention calculation in downsampled spaces and employs cascaded modulation operation to pursue omni-kernel feature refinement and high-dimensional representation learning. As such, the model achieves a better tradeoff between complexity and accuracy.

friendly, showing competitive performance on par with Transformer counterparts. Inspired by this modulation technique, we acquire the approximate omni-kernel feature modeling ability by integrating the Transformer layer (Figure 1 (a)) and modulation design (Figure 1 (b)) within a block. As illustrated in Figure 1 (c), the context branch (CTX) is implemented through a Transformer block at a downsampled scale, which retains the ability of SA to model global features while striking a trade-off between complexity and accuracy. The local and mesoscale receptive fields are complemented by modulating the result of SA in series using depth-wise convolutions of different kernel sizes. Compared to the canonical modulation design, our block provides real context modeling and performs cascaded modulation processes, mapping input features into higher-dimensional feature spaces. Additionally, our context branch is content-aware, which is beneficial for dealing with spatially varying degradations. Moreover, we explore a bioinspired parameter-sharing mechanism that shares parameters across different attention layers, improving both efficiency and performance.

Additionally, we present a dual-domain feed-forward network (DFFN) to improve dual-domain representation learning. Specifically, DFFN first utilizes GEGLU (Shazeer, 2020) to achieve spatial-domain signal interactions. Subsequently, the resulting features pass through the fast Fourier transform (FFT) to obtain the spectra, which are then modulated by the learnable parameters and transformed back to the spatial domain through the inverse IFFT. Next, the results interact with spatial features under the guidance of attention weights. By doing these, our DFFN achieves intra- and inter-domain interactions, improving the representational ability.

The unit of our U-shaped Modumer is built upon the above modulation-based SA block and DFFN. Unlike other Transformer-based restoration algorithms that utilize a uniform block throughout the model, we adopt a channel-wise modulation-based SA block at the initial scale to enable more efficient global feature modeling. For lower-resolution features at deeper scales, we apply spatial-wise blocks, effectively capturing spatial details. Based on these designs, Modumer achieves state-of-the-art performance on several image restoration tasks with lower complexity and fewer parameters (see Figure 2). For deraining, Modumer outperforms the previous state-of-the-art method (Zhou et al., 2024a) by 0.73 dB on AGAN-Data (Qian et al., 2018). For motion blur removal, Modumer significantly surpasses other algorithms on the HIDE dataset (Shen et al., 2019), displaying its strong capability of deblurring. Modumer also exhibits the potential on the CSD (Chen et al., 2021) dataset for the desnowing task and is superior to the previous best model (Cui et al., 2024a) by 0.74 dB in terms of PSNR. Also, on the Haze4k (Liu et al., 2021b) dataset for dehazing, it obtains 34.69 dB PSNR, an improvement of 0.54 dB over the previous state-of-the-art method (Cui et al., 2024a).

To summarize, the main contributions of this study are listed as follows:

- We present a novel attention block that consecutively modulates the self-attention results from downsampled features, providing efficient omni-kernel modulation and high-dimensional representational capability. A bioinspired parameter-sharing mechanism is introduced to improve both efficiency and performance.

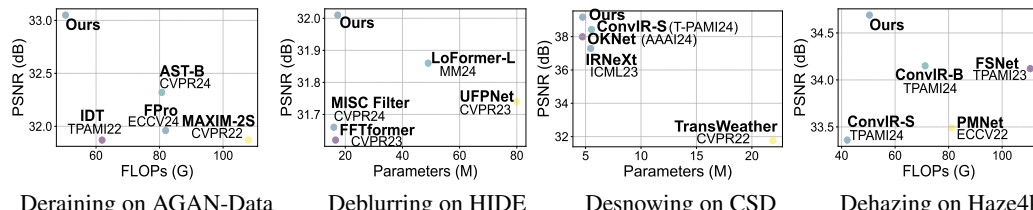

| Deraining on AGAN-Data | Deblurring on HIDE | Desnowing on CSD | Dehazing on Haze4k |

Figure 2: Computation comparisons between the proposed model and state-of-the-art algorithms on AGAN-Data (Qian et al., 2018), HIDE (Shen et al., 2019), CSD Chen et al. (2021), and Haze4k (Liu et al., 2021b) for deraining, motion deblurring, desnowing, and dehazing, respectively.

- We develop a dual-domain feed-forward network that achieves spatial-spatial and spectral-spatial interactions.

- We deploy channel-wise Transformer blocks at the first scale while using spatial-wise blocks at deeper scales with lower-restoration features, resulting in our effective and efficient image restoration network, dubbed Modumer.

- Extensive experiments show that Modumer achieves state-of-the-art performance on ten benchmark datasets for five representative image restoration tasks, including image motion deblurring, image deraining, image dehazing, image desnowing, and low-light image enhancement. Moreover, Modumer produces promising performance in all-in-one scenarios.

## 2 RELATED WORKS

### 2.1 IMAGE RESTORATION

As a fundamental vision task, image restoration aims to reconstruct a sharp image from a degraded observation (Cho et al., 2021; Ruan et al., 2022). To resolve this heavily ill-posed problem, many conventional algorithms have been proposed based on hand-crafted features and assumptions to reduce the solution space (He et al., 2010). Recently, deep learning methods have remarkably boosted the performance of various image restoration tasks by learning generalizable features from large-scale collected data. These methods can be roughly divided into CNN-based and Transformer-based categories. CNN-based methods leverage attention mechanisms to attend to informative information for different dimensions (Qin et al., 2020; Zamir et al., 2021; Cui et al., 2023a), *e.g.,* pixel, spatial, and channel. Also, they employ advanced techniques to enlarge the receptive fields and model multi-scale features (Son et al., 2021; Liu et al., 2020; Nah et al., 2017; Jiang et al., 2020; Cui et al., 2023c), such as the encoder-decoder architecture, atrous convolution, and multi-stage learning strategy. Subsequently, Transformer methods scale the receptive field to global features via the SA layer (Tsai et al., 2022; Guo et al., 2022). To enhance its efficiency on low-level vision tasks, a few algorithms confine the SA region to fixed windows or strips (Wang et al., 2022; Liang et al., 2021; Song et al., 2022), which impedes the inherent potential of SA. Moreover, they cannot model multi-scale features within a single unit, limiting their capability for removing degradations of different sizes. In this paper, we apply SA to downsampled embedding spaces to capture global dependencies and use the cascaded modulation operation to complement the missing local information.

### 2.2 MODULATION DESIGN

The modulation mechanism (Ma et al., 2024b; Guo et al., 2023a) considers context modeling using a large-kernel convolutional unit and modulates the projected inputs using element-wise multiplication, which has exhibited cutting-edge performance in high-level vision tasks. FocalNet (Yang et al., 2022) utilizes a stack of depth-wise convolutional layers to implement hierarchical contextualization and uses gated aggregation to selectively gather contexts. Afterward, EfficientMod (Ma et al., 2024b) adopts a simpler method for context modeling using a series of linear projections and depth-wise convolution. MambaOut (Yu & Wang, 2024) and Conv2former (Hou et al., 2024) use $7 \times 7$ depth-wise convolutions to extract contextual features. Recently, StarNet (Ma et al., 2024a) uncovers that the strong representational capacity of element-wise multiplication originates from implicitly high-dimensional spaces. However, the receptive fields of the context branch in these methods are limited. In contrast, our method involves long-range contextual signals by applying SA to downsampled embedding spaces, striking a balance between complexity and accuracy. Moreover,

Figure 3: The network architecture of our U-shaped Modumer. We employ channel-wise modulation block (CMB) with shared parameters at the first scale while using spatial-wise modulation block (SMB) at deeper scales which involve lower-resolution features. This can strike a better balance between the complexity and the representational ability. The DFFN enhances dual-domain frequency learning via spatial-spatial and spatial-spectral interactions.

the mesoscale and local information is used to modulate the SA results via cascaded modulation, achieving omni-kernel refinement and mapping inputs into higher-dimensional spaces.

## 3 METHODOLOGY

In this section, we first introduce the overall architecture of Modumer. Subsequently, the proposed components are delineated individually, including two kinds of attention layers (CMB, SMB), the parameter-sharing mechanism, and the dual-domain feed-forward network (DFFN).

### 3.1 OVERALL PIPELINE

Modumer follows the encoder-decoder design (see Figure 3). We employ a channel-wise modulation block (CMB) at the first scale, as the channel-wise SA can implicitly capture the large-range features efficiently while using a spatial-wise modulation block (SMB) in the other two lower-resolution scales. As such, the model strikes a better balance between complexity and representational capacity.

Specifically, given an image, we use a $3 \times 3$ convolution to extract the embedding features of size $\mathbb{R}^{C \times H \times W}$, where $C$ denotes the channel count while $H \times W$ defines the spatial index. Subsequently, the features are fed into the three-scale encoder sub-network to produce the in-depth features. Each scale contains several Transformer blocks, whose calculation process is formulated as

$$\mathbf{X}'_k = \text{CMB/SMB}(\mathbf{X}_{k-1}) + \mathbf{X}_{k-1}, \tag{1}$$

$$\mathbf{X}_k = \text{DFFN}(\mathbf{X}'_k) + \mathbf{X}'_k, \tag{2}$$

where $\mathbf{X}_{k-1}$ and $\mathbf{X}_k$ are the output of the last and current Transformer block, respectively. In the encoder stage, the resolution of the features is gradually downsampled using *bilinear* interpolation while the channel capability is doubled using a $3 \times 3$ convolution. Next, the in-depth features pass through the symmetric decoder network to generate the clean features. In this process, the resolution of features is progressively restored to the original size using *bilinear* interpolation and $3 \times 3$ convolution. Meanwhile, the skip connection is adopted to combine the encoder and decoder features via concatenation. The yielded features after the three-level decoder are finally processed by a refinement stage involving $r$ Transformer blocks and a $3 \times 3$ convolution to generate the residual image, which is added to the original input image to obtain the model output. Next, we present the internal components of the Transformer block.

### 3.2 CHANNEL-WISE MODULATION BLOCK (CMB)

The architectural details of CMB are illustrated in Figure 4 (a). CMB contains a downsampled channel-wise SA layer for global information modeling and two depth-wise convolutional branches modulating the SA result to complement local and mesoscale receptive fields and map features into higher-dimensional spaces. The calculation process of CMB can be formally expressed as

$$\hat{\mathbf{X}}_{\text{CMB}} = W_2 \left( \hat{\mathbf{X}}_{M7 \times 7} \left( W_1 (\hat{\mathbf{X}}_{M3 \times 3} \odot \hat{\mathbf{X}}_{\text{D-CSA}}) \right) \right), \tag{3}$$

(a) CMB                                      (b) SMB

Figure 4: Module architectures of channel and spatial modulation blocks (CMB‖SMB).

where $\hat{\mathbf{X}}_{\mathrm{CMB}}$, $\mathbf{X}_{\mathrm{CMB}}$ denote the output and input of CMB, respectively. D-CSA is a downsampled channel-wise self-attention layer. $\hat{\mathbf{X}}_{Mn\times n}$ is the modulation branch with the kernel size of $n \times n$, encoding local information. $W_1$ and $W_2$ are $1 \times 1$ convolutions for refinement.

**D-CSA.** Compared to the normal channel SA, our version computes attention maps in a downsample space, resulting in high efficiency. We assume that the number of heads is 1 and consider D-CSA as a single-head fashion. Given the normalized input $\mathbf{X}_{\mathrm{N}} \in \mathbb{R}^{C \times H \times W}$, D-CSA first utilizes the projection layers to produce query, key, and value tensors by $\mathbf{Q} = W_Q \mathbf{X}_{\mathrm{N}}$, $\mathbf{K} = W_K \mathbf{X}_{\mathrm{N}}$, and $\mathbf{V} = W_V \mathbf{X}_{\mathrm{N}}$, where $W_{(\cdot)}$ denotes parameters of $1 \times 1$ point-wise convolution. Then, the obtained $\mathbf{Q}, \mathbf{K}, \mathbf{V}$ tensors are reshaped into the size of $C \times N$, $N \times C$, and $C \times N$, respectively, where $N = H \times W$. The query and key tensors are further normalized and downsampled to prepare for cross-covariance attention. The transposed attention map is calculated by $\mathbf{Q}$ and $K$ with size of $\mathbb{R}^{C \times C}$. The output of D-CSA can be obtained by

$$\hat{\mathbf{X}}_{\mathrm{D-CSA}} = \mathrm{Softmax}(\mathbf{Q}\mathbf{K}/\tau)\mathbf{V}, \tag{4}$$

where $\tau$ is a learnable temperature parameter and $\hat{\mathbf{X}}_{\mathrm{D-CSA}} \in \mathbb{R}^{C \times N}$ is reshaped to the original input feature size of $\mathbb{R}^{H \times W \times C}$ for further modulation operation.

**Modulation design.** D-CSA encodes downsampled global information while ignoring the fine-grained local details when downsampling features. To complement local information, we first filter the initially generated $\mathbf{V}$ tensor using a $3 \times 3$ depth-wise convolution, as $\mathbf{V}$ has been refined by the convolutional layer. This process is expressed as

$$\hat{\mathbf{X}}_{M3\times 3} = \mathrm{Sigmoid}(Dw_{3\times 3}(\mathbf{V})) \odot \mathbf{V}, \tag{5}$$

where $Dw_{3\times 3}$ is a depth-wise convolution of kernel size $3 \times 3$. Next, we modulate the output of D-CSA with the locally filtered result via element-wise multiplication. By doing this, the model can capture downsampled global and local dependencies and map inputs into high-dimensional spaces to improve the representational capability. To simplify the analyses, assuming the scenario involves a single-pixel input $x \in \mathbb{R}^{d \times 1}$ and a single-element output, $\hat{x} \in \mathbb{R}^{1 \times 1}$, where $d$ is the channel count, we define $w_1, w_2 \in \mathbb{R}^{1 \times d}$ as convolution parameters. The modulation process involving a single convolution within each branch can be written as

$$w_1^\top x \odot w_2^\top x = \left(\sum_{i=1}^{d} w_1^i x^i\right) \odot \left(\sum_{j=1}^{d} w_2^j x^j\right) \tag{6}$$

$$= \sum_{i=1}^{d}\sum_{j=1}^{d} w_1^i w_2^j x^i x^j \tag{7}$$

$$= \underbrace{\frac{\alpha_{1,1} x^1 x^1 + \cdots + \alpha_{2,3} x^2 x^3 + \cdots + \alpha_{d,d} x^d x^d}{d(d+1)/2}}, \quad \alpha_{i,j} = \begin{cases} w_1^i w_2^j, & i = j, \\ w_1^i w_2^j + w_1^j w_2^i & i \neq j. \end{cases} \tag{8}$$

where $i, j$ index the channel. We can observe that each item in Eq. 8 presents a non-linear association with $x$ and is an individual dimension, indicating that this case achieves a representation in a $d(d+1)/2$ implicit dimensional feature space. Note that besides convolutions, the branches in our modulation design experience complicated SA, further improving the representational capability.

Additionally, we apply a $7 \times 7$ kernel branch to further modulate the preceding outcome and supply mesoscale receptive fields.

**Parameter sharing.** Inspired by the relationship between the hippocampus and cortex in the brain (Whittington et al., 2020; 2021), where different regions and layers of the cortex, despite performing different tasks, all receive and send information from a single shared memory in the hippocampus, we consider the attention layer as the hippocampus while the feed-forward layer as the cortex, forming our parameter-sharing mechanism illustrated in the left part of Figure 3. Interestingly, this design not only saves parameters but also improves the performance. More discussions are provided in the Appendix.

### 3.3 SPATIAL-WISE MODULATION BLOCK (SMB)

Figure 4 (b) presents the details of SMB, which mainly has three branches: a downsampled spatial-wise attention unit (D-SSA), and two modulation operators. The output of SMB is obtained by

$$\hat{\mathbf{X}}_{\text{SMB}} = W_4 \left( \hat{\mathbf{X}}_{M7\times7} \left( W_3(\hat{\mathbf{X}}_{M3\times3} \odot \hat{\mathbf{X}}_{\text{D-SSA}}) \right) \right), \tag{9}$$

where $\hat{\mathbf{X}}_{\text{D-SSA}}$ is the outcome of D-SSA.

**D-SSA.** D-SSA is used in low-resolution scales to model spatial global features. Similarly, we also assume the number of heads is 1 to transfer D-SSA to single-head mode. Given any input $\mathbf{X} \in \mathbb{R}^{H \times W \times C}$, it is first processed by the layer normalization to yield $\mathbf{X}_{\text{N}}$. Then, query ($\mathbf{Q}$), key ($\mathbf{K}$), and value ($\mathbf{V}$) tensors are produced by $\mathbf{Q} = W^Q \mathbf{X}_{\text{N}}$, $\mathbf{K} = W^K \mathbf{X}_{\text{N}} \downarrow$, and $\mathbf{V} = W^V \mathbf{X}_{\text{N}} \downarrow$, where $\mathbf{K}$ and $\mathbf{V}$ are generated from the downsampled input ($\mathbf{X}_{\text{N}} \downarrow$) for high efficiency. After reshaping $\mathbf{Q}$, $\mathbf{K}$, and $\mathbf{V}$ to new tensors of size $N \times C$, $C \times N'$, $N' \times C$, respectively, where $N = H \times W$ and $N' = H/8 \times W/8$, the calculation process of D-SSA is formulated as

$$\hat{\mathbf{X}}_{\text{D-SSA}} = \text{Softmax}(\frac{\mathbf{QK}}{\sqrt{C}})\mathbf{V}. \tag{10}$$

**Modulation design.** Similar to CMB, we utilize a cascaded modulation design with kernel sizes of $3 \times 3$ and $7 \times 7$ to complement local and mesoscale information. As such, the model is equipped with an approximate omni-kernel modulation ability, *i.e.,* local-mesoscale-global.

### 3.4 DUAL-DOMAIN FEED-FORWARD NETWORK (DFFN)

DFFN facilitates the spatial-spatial and spatial-spectral interactions for high-fidelity reconstruction. Figure 3 illustrates the architecture. To be specific, given input features $\mathbf{X} \in \mathbb{R}^{H \times W \times C}$, after the layer normalization, DFFN first performs GEGLU (Shazeer, 2020) as

$$\hat{\mathbf{X}}_{\text{S-S}} = W_7 \left( \text{GELU} \left( Dw_3^1 W_5(\mathbf{X}_{\text{N}}) \right) \odot Dw_3^2 W_6(\mathbf{X}_{\text{N}}) \right) \tag{11}$$

where $W_5$, $W_6$ and $W_7$ denote $1 \times 1$ convolutions. $Dw_3^1$ and $Dw_3^2$ are $3 \times 3$ depth-wise convolutions. $\mathbf{X}_{\text{N}}$ is the normalized input and $\hat{\mathbf{X}}_{\text{S-S}}$ is the spatial-spatial interaction output.

Furthermore, DFFN conducts spatial-spectral interactions by adding the Fourier-domain refined result and spatial features together under the guidance of learnable attention weights. The calculation process can be formulated as

$$\hat{\mathbf{X}}_{\text{DFFN}} = \alpha \mathbf{X}_{\text{Spectral}} + (1 - \alpha)\hat{\mathbf{X}}_{\text{S-S}} \tag{12}$$

$$\mathbf{X}_{\text{Spectral}} = \mathcal{P}^{-1} \left( \mathcal{F}^{-1} \left( W \odot \left( \mathcal{F}(\mathcal{P}(\hat{\mathbf{X}}_{\text{S-S}})) \right) \right) \right) \tag{13}$$

where $\mathcal{F}$ and $\mathcal{F}^{-1}$ denote the fast Fourier transform and the inverse transform, respectively. $\mathcal{P}$ and $\mathcal{P}^{-1}$ are windows partition operation and the inverse transformation, respectively. $W$ is the learnable parameter to filter the frequency signals. $\alpha$ is the learnable parameter to control dual-domain information aggregation.

## 4 EXPERIMENTS

To validate the efficacy of the proposed Modumer, we evaluate the model on two kinds of tasks, general image restoration and all-in-one image restoration. The former trains different model copies for different datasets while the latter uses a single model for different degradation types and levels. In this section, we first present the implementation details, experimental results, and ablation studies for general image restoration. Subsequently, we apply our model to the all-in-one settings.

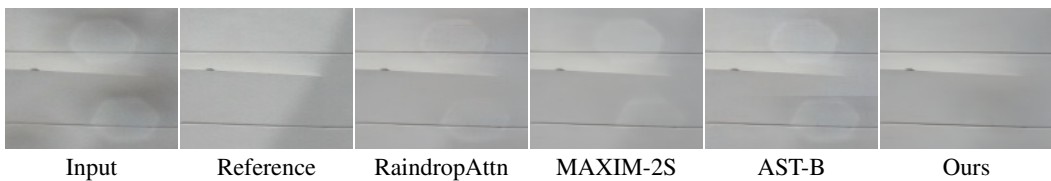

| Input | Reference | RaindropAttn | MAXIM-2S | AST-B | Ours |

Figure 5: Visual comparisons on the raindrop AGAN-Data (Qian et al., 2018) dataset.

Table 1: The dataset summary for five tasks under general image restoration.

| Task | Deraining | Motion deblurring | Dehazing | Desnowing | Low-light image enhancement |
|---|---|---|---|---|---|
| Dataset | SPAD‖AGAN-Data | GoPro‖HIDE | Haze4k‖GTA5 | CSD‖SRRS‖Snow100K | LOL-v2 |

Table 2: Quantitative comparisons on AGAN-Data (Qian et al., 2018) for raindrop removal.

| Methods | PSNR | SSIM |
|---|---|---|
| Uformer (Wang et al., 2022) | 29.42 | 0.906 |
| TransWeather (Valanarasu et al., 2022) | 30.17 | 0.916 |
| Quan *et al.* (Quan et al., 2019) | 31.37 | 0.918 |
| AttenGAN (Qian et al., 2018) | 31.59 | 0.917 |
| IDT (Xiao et al., 2022) | 31.87 | 0.931 |
| MAXIM-2S (Tu et al., 2022) | 31.87 | 0.935 |
| AWRCP (Ye et al., 2023) | 31.93 | 0.931 |
| FPro (Zhou et al., 2024b) | 31.96 | 0.937 |
| AST-B (Zhou et al., 2024a) | 32.32 | 0.935 |
| **Ours-S** | **33.05** | **0.946** |

Table 3: Quantitative results on SPAD (Wang et al., 2019) for rain streak removal.

| Methods | PSNR | SSIM |
|---|---|---|
| SEIDNet (Lin et al., 2022) | 44.96 | 0.9911 |
| Fu *et al.* (Fu et al., 2023) | 45.03 | 0.9907 |
| Restormer (Zamir et al., 2022a) | 46.25 | 0.9911 |
| SCD-Former (Guo et al., 2023b) | 46.89 | 0.9941 |
| IDT (Xiao et al., 2022) | 47.34 | 0.9929 |
| Uformer Wang et al. (2022) | 47.84 | 0.9925 |
| DRSformer (Chen et al., 2023b) | 48.53 | 0.9924 |
| FPro (Zhou et al., 2024b) | 48.99 | 0.9936 |
| AST-B (Zhou et al., 2024a) | 49.51 | **0.9942** |
| **Ours-S** | **49.57** | **0.9942** |

## 4.1 GENERAL IMAGE RESTORATION

### 4.1.1 IMPLEMENTATION DETAILS

We evaluate our model on five representative tasks with **ten** benchmark datasets. The used datasets are summarized in Table 1. We adopt the dual-domain loss functions (Cho et al., 2021; Kong et al., 2023; Cui et al., 2023a) to train the network for 300,000 iterations with the Adam optimizer. The deblurring task needs another 300,000 iterations following (Kong et al., 2023). The initial learning is set to $1e^{-3}$, which is gradually reduced to $1e^{-7}$ with the cosine annealing strategy. The patch size is set to $128 \times 128$ and the batch size is 32. We adopt the same data augmentation strategy as (Zamir et al., 2022a). The window size in DFFN and the downsampling ratio in SA are set to 8. According to the complexity of different datasets, we present two model versions, Modumer-S (small) and Modumer-B (base). For Modumer-S, we set the channel count to 42, and $[L_1, L_2, L_3, L_r]$ as [2,2,4,4], while for the base model, we set the channel number to 48, and [6,6,13,4] for $[L_1, L_2, L_3, L_r]$. FLOPs are measured on $3 \times 256 \times 256$ patches. Due to the space limit, image enhancement results and more visualizations are presented in the Appendix. In tables, the best results are **highlighted**.

### 4.1.2 RESULTS

**Image deraining.** The numerical results on the raindrop dataset AGAN-Data (Qian et al., 2018) are presented in Table 2. Our method significantly outperforms the recent Transformer-based AST-B (Zhou et al., 2024a) and FPro (Zhou et al., 2024b) by 0.73 dB and 1.09 dB, respectively, while consuming lower complexity, as illustrated in Figure 2 (a). Figure 5 shows that our method is more effective in raindrop removal than competitors. Moreover, the comparison results on the rain streak dataset SPAD (Wang et al., 2019) are reported in Table 3. As seen, our method achieves the best performance in terms of PSNR, outperforming the previous state-of-the-art algorithm (Zhou et al., 2024a) by 0.06 dB PSNR.

**Image motion deblurring.** We conduct experiments for motion deblurring on the GoPro (Nah et al., 2017) dataset and compare our results with state-of-the-art works in Table 4. Our method significantly surpasses the recent frequency-based Transformer model (Mao et al., 2024) by 0.18 dB PSNR while using 65% fewer parameters. Compared to the recent convolutional network ConvIR-L (Cui et al., 2024a), our method achieves a notable gain of 0.99 dB PSNR with comparable parameters and FLOPs. The visual results in Figure 6 show that our model recovers more structural details from

Table 4: Image motion deblurring results. Our model is trained only on the GoPro Nah et al. (2017) dataset and directly applied to the GoPro (Nah et al., 2017) and HIDE Shen et al. (2019) datasets.

| Methods | GoPro | | HIDE | | Params (M) | FLOPs (G) |
|---|---|---|---|---|---|---|
| | PSNR | SSIM | PSNR | SSIM | | |
| DMPHN (Zhang et al., 2019a) | 31.20 | 0.940 | 29.09 | 0.924 | - | - |
| DBGAN (Zhang et al., 2020) | 31.10 | 0.942 | 28.94 | 0.915 | 11.6 | 760 |
| Restormer (Zamir et al., 2022a) | 32.92 | 0.961 | 31.22 | 0.942 | 26.1 | 135 |
| Stripformer (Tsai et al., 2022) | 33.08 | 0.962 | 31.03 | 0.940 | 20.0 | 170 |
| GRL (Li et al., 2023) | 33.93 | 0.968 | 31.65 | 0.947 | 20.2 | 1289 |
| UFPNet (Fang et al., 2023) | 34.06 | 0.968 | 31.74 | 0.947 | 80.3 | 243 |
| FSNet (Cui et al., 2023b) | 33.29 | 0.963 | 31.05 | 0.941 | 13.28 | 111 |
| FFTformer (Kong et al., 2023) | 34.21 | **0.969** | 31.62 | 0.946 | 16.6 | 131 |
| ConvIR-L (Cui et al., 2024a) | 33.28 | 0.963 | - | - | 14.83 | 129 |
| MLWNet-B (Gao et al., 2024) | 33.83 | 0.968 | - | - | - | 108 |
| MISC Filter (Liu et al., 2024) | 34.10 | **0.969** | 31.66 | 0.946 | 16.0 | - |
| LoFormer-L (Mao et al., 2024) | 34.09 | **0.969** | 31.86 | **0.949** | 49.0 | 126 |
| **Ours-B** | **34.27** | **0.969** | **32.01** | **0.949** | 17.35 | 139 |

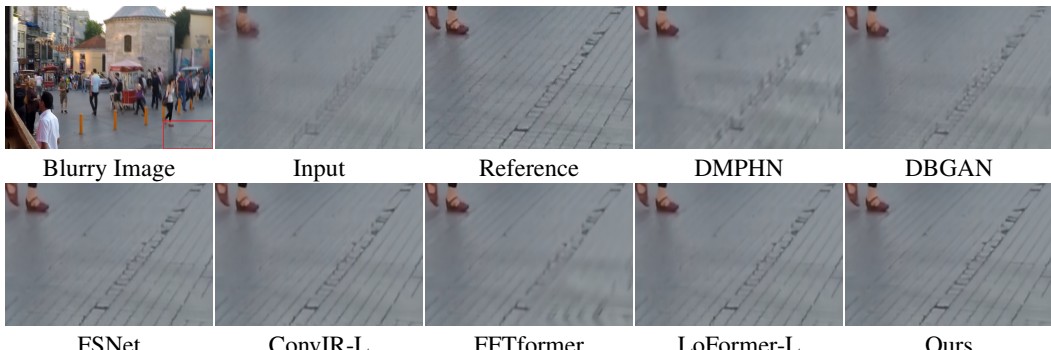

| Blurry Image | Input | Reference | DMPHN | DBGAN |
|---|---|---|---|---|
| FSNet | ConvIR-L | FFTformer | LoFormer-L | Ours |

Figure 6: Deblurred results on the GoPro (Nah et al., 2017) dataset. Compared to other algorithms, the proposed method restores more details and clearer structures from the input.

Table 5: Image dehazing comparisons on the Haze4k (Liu et al., 2021b) dataset.

| Methods | PSNR | SSIM |
|---|---|---|
| MSBDN (Dong et al., 2020a) | 22.99 | 0.85 |
| FFA-Net (Qin et al., 2020) | 26.96 | 0.95 |
| DMT-Net (Liu et al., 2021c) | 28.53 | 0.96 |
| PMNet (Ye et al., 2022) | 33.49 | 0.98 |
| FSNet (Cui et al., 2023b) | 34.12 | **0.99** |
| ConvIR-S (Cui et al., 2024a) | 33.36 | **0.99** |
| ConvIR-B (Cui et al., 2024a) | 34.15 | **0.99** |
| **Ours-S** | **34.69** | **0.99** |

Table 6: Quantitative results on GTA5 (Yan et al., 2020) for night haze removal.

| Methods | PSNR | SSIM |
|---|---|---|
| MRP (Zhang et al., 2017) | 20.92 | 0.646 |
| Ancuti *et al.* Ancuti et al. (2016) | 20.59 | 0.623 |
| CycleGAN (Engin et al., 2018) | 21.75 | 0.696 |
| Yan *et al.* (Yan et al., 2020) | 27.00 | 0.850 |
| Jin *et al.* Jin et al. (2023) | 30.38 | 0.904 |
| ConvIR-S Cui et al. (2024a) | 31.68 | 0.917 |
| ConvIR-B Cui et al. (2024a) | 31.83 | 0.921 |
| **Ours-S** | **32.04** | **0.928** |

the hard example. We further apply our model pre-trained on GoPro to the HIDE (Shen et al., 2019) dataset. The quantitative results presented in Table 4 show that our method obtains the best result in PSNR with a prominent gain of 0.15 dB over the second-best LoFormer-L (Mao et al., 2024), demonstrating the better generalization ability of our model.

**Image dehazing.** We perform dehazing experiments on the Haze4k (Liu et al., 2021b) dataset. The numerical results are presented in Table 5. Our model attains a significant performance gain of 0.54 dB PSNR over the recent algorighm (Cui et al., 2024a) with lower FLOPs, as illustrated in Figure 2 (d). Compared to the CNN-based method FSNet (Cui et al., 2023b), our advantage is more obvious with much lower complexity. Figure 7 shows that our model can better deal with haze degradations than other algorithms. Additionally, we provide comparison results on a nighttime dehazing dataset GTA5 (Yan et al., 2020) in Table 6. Our Modumer-S is still superior to the strong competitors.

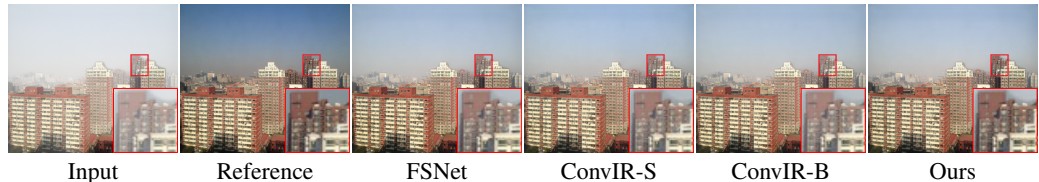

Figure 7: Image dehazing comparisons on the Haze4k (Liu et al., 2021b) dataset.

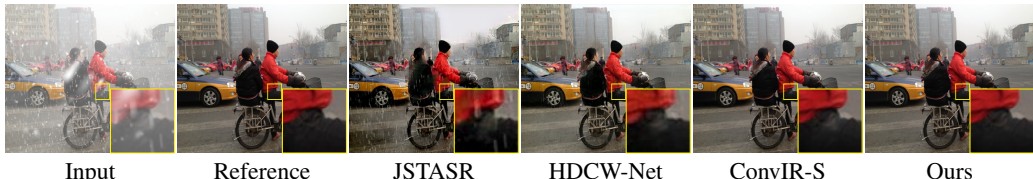

Figure 8: Image desnowing comparisons on the CSD Chen et al. (2021) dataset.

Table 7: Image desnowing comparisons on three widely-used datasets: CSD (Chen et al., 2021), SRRS (Chen et al., 2020), and Snow100K (Liu et al., 2018).

| Methods | CSD | | SRRS | | Snow100K | | Params | FLOPs |
|---|---|---|---|---|---|---|---|---|
| | PSNR | SSIM | PSNR | SSIM | PSNR | SSIM | (M) | (G) |
| DesnowNet (Liu et al., 2018) | 20.13 | 0.81 | 20.38 | 0.84 | 30.50 | 0.94 | 15.6 | 1.7K |
| JSTASR (Chen et al., 2020) | 27.96 | 0.88 | 25.82 | 0.89 | 23.12 | 0.86 | 65 | - |
| HDCW-Net (Chen et al., 2021) | 29.06 | 0.91 | 27.78 | 0.92 | 31.54 | 0.95 | 6.99 | 9.78 |
| SMGARN (Cheng et al., 2022) | 31.93 | 0.95 | 29.14 | 0.94 | 31.92 | 0.93 | 6.86 | 450.3 |
| TransWeather (Valanarasu et al., 2022) | 31.76 | 0.93 | 28.29 | 0.92 | 31.82 | 0.93 | 21.9 | 5.64 |
| MSP-Former (Chen et al., 2023a) | 33.75 | 0.96 | 30.76 | 0.95 | 33.43 | **0.96** | 2.83 | 4.42 |
| OKNet (Cui et al., 2024b) | 37.99 | 0.99 | 31.70 | **0.98** | 33.75 | 0.95 | 4.72 | 39.67 |
| IRNeXt (Cui et al., 2023c) | 37.29 | **0.99** | 31.91 | **0.98** | 33.61 | 0.95 | 5.46 | 42.09 |
| ConvIR-S (Cui et al., 2024a) | 38.43 | **0.99** | 32.25 | **0.98** | 33.79 | 0.95 | 5.53 | 42.1 |
| **Ours-S** | **39.17** | **0.99** | **32.48** | **0.98** | **34.58** | **0.96** | 4.74 | 50.39 |

**Image desnowing.** Furthermore, we verify the effectiveness of our model in snow removal using three datasets: CSD (Chen et al., 2021), SRRS Chen et al. (2020), and Snow100K Liu et al. (2018). The quantitative results are presented in Table 7. With similar computation overhead, our method achieves 39.17 dB PSNR on the CSD dataset, 0.74 dB higher than the second-best algorithm (Cui et al., 2024a). The superiority of our model can also be found on the other two datasets, demonstrating the effectiveness of our model in snow removal. Figure 8 shows that our model yields a more favorable image by removing more snow degradations.

### 4.1.3 ABLATION STUDIES

We perform ablation studies by training our small model for 70,000 iterations on GoPro (Nah et al., 2017). More ablation results can be found in the Appendix.

Table 8 shows the results of individually removing the proposed component from the complete model. Removing

Table 8: Ablation studies for each component.

| Mod. $3 \times 3$ | Mod. $7 \times 7$ | Sharing | DFFN | PSNR | Params. |
|---|---|---|---|---|---|
| | ✓ | ✓ | ✓ | 31.70 | 4.72M |
| ✓ | | ✓ | ✓ | 31.69 | 4.64M |
| ✓ | ✓ | | ✓ | 31.78 | 4.79M |
| ✓ | ✓ | ✓ | | 31.69 | 4.74M |
| ✓ | ✓ | ✓ | ✓ | 31.82 | 4.74M |

our modulation branch leads to degraded performance compared to the full model. Our parameter-sharing mechanism achieves 0.04 dB PSNR performance improvement while consuming fewer parameters. Employing only the spatial-spatial interactions, *i.e.,* GEGLU, in the feed-forward network achieves 31.69 dB PSNR, which is 0.13 dB lower than our full model. These results demonstrate the effectiveness of our proposed modules and mechanism.

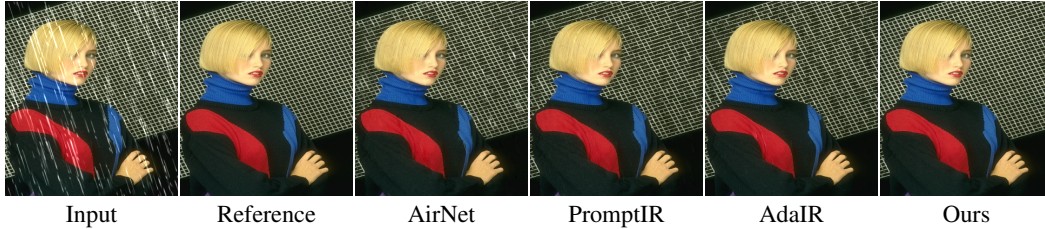

| Input | Reference | AirNet | PromptIR | AdaIR | Ours |

Figure 9: Visual comparisons on the Rain100 (Yang et al., 2017) dataset under the all-in-one setting. The image produced by our model is closer to the reference image, such as the background regions.

Table 9: Dataset summary for all-in-one image restoration. Motion deblurring and low-light enhancement are only used for the five-task setting.

| Task | Desnoiwing | Dehazing | Deraining | Motion deblurring | Low-light enhancement |
|---|---|---|---|---|---|
| **Train** | BSD400∥WED | RESIDE | Rain100L | GoPro | LOL-v1 |
| **Test** | BSD68 | SOTS-Outdoor | Rain100L | GoPro | LOL-v1 |

Table 10: Quantitative comparisons on three image restoration tasks under the all-in-one setting.

| | Denoising on BSD68 | | | | | | Deraining on Rain100L | | Dehazing on SOTS | | Average | |
| | $\sigma = 15$ | | $\sigma = 25$ | | $\sigma = 50$ | | | | | | | |
| Method | PSNR | SSIM | PSNR | SSIM | PSNR | SSIM | PSNR | SSIM | PSNR | SSIM | PSNR | SSIM |
|---|---|---|---|---|---|---|---|---|---|---|---|---|
| BRDNet (Tian et al., 2020) | 32.26 | 0.898 | 29.76 | 0.836 | 26.34 | 0.693 | 27.42 | 0.895 | 23.23 | 0.895 | 27.80 | 0.843 |
| LPNet (Gao et al., 2019) | 26.47 | 0.778 | 24.77 | 0.748 | 21.26 | 0.552 | 24.88 | 0.784 | 20.84 | 0.828 | 23.64 | 0.738 |
| FDGAN (Dong et al., 2020b) | 30.25 | 0.910 | 28.81 | 0.868 | 26.43 | 0.776 | 29.89 | 0.933 | 24.71 | 0.929 | 28.02 | 0.883 |
| MPRNet (Zamir et al., 2021) | 33.54 | 0.927 | 30.89 | 0.880 | 27.56 | 0.779 | 33.57 | 0.954 | 25.28 | 0.955 | 30.17 | 0.899 |
| DL (Fan et al., 2019) | 33.05 | 0.914 | 30.41 | 0.861 | 26.90 | 0.740 | 32.62 | 0.931 | 26.92 | 0.931 | 29.98 | 0.876 |
| AirNet (Li et al., 2022) | 33.92 | 0.933 | 31.26 | 0.888 | 28.00 | 0.797 | 34.90 | 0.968 | 27.94 | 0.962 | 31.20 | 0.910 |
| PromptIR (Potlapalli et al., 2023) | 33.98 | 0.933 | 31.31 | 0.888 | 28.06 | 0.799 | 36.37 | 0.972 | 30.58 | 0.974 | 32.06 | 0.913 |
| AdaIR (Cui et al., 2024c) | 34.12 | 0.935 | 31.45 | 0.892 | 28.19 | 0.802 | 38.64 | 0.983 | 31.06 | **0.980** | 32.69 | 0.918 |
| **Ours** | 34.15 | 0.936 | 31.50 | 0.893 | 28.25 | 0.805 | 38.78 | 0.984 | 31.17 | 0.979 | **32.77** | **0.919** |

## 4.2 ALL-IN-ONE IMAGE RESTORATION

### 4.2.1 IMPLEMENTATION DETAILS

Following the recent algorithm (Potlapalli et al., 2023; Cui et al., 2024c), we perform all-in-one experiments under three-task and five-task settings with Modumer-B. The dataset summary is presented in Table 9. The model is trained on 32 samples of size $128 \times 128$ in an iteration with a learning rate of $2e^{-4}$ using Adam. The models are trained for 150 epochs with $L_1$ loss function.

### 4.2.2 RESULTS

For the three-task setting, the model is trained on the mixed datasets obtained from denoising, dehazing, and deraining. Table 10 shows that our model achieves an average score of 32.77 dB PSNR, 0.08 dB higher than the recent frequency-based AdaIR (Cui et al., 2024c). Moreover, our method attains the best performance on most metrics. Particularly on the deraining problem, a 0.14 dB performance gain is produced by our model over AdaIR. Figure 9 demonstrates that our model is more effective in removing rain streaks, resulting in a noticeably cleaner image. We provide the result for the five-task scenario in the Appendix.

## 5 CONCLUSION

This study presents an effective and efficient Transformer model for image restoration, termed Modumer. The model incorporates the different downsampled self-attention layers with cascaded modulation designs, which can model omni-receptive field features, obtain a better balance between complexity and accuracy, and map features into high-dimensional spaces. Moreover, we investigate a bioinspired parameter-sharing mechanism in attention layers, improving efficiency and performance. In addition, we introduce a feed-forward network to facilitate intra- and inter-domain interactions. Extensive experimental results on ten datasets for general image restoration and two all-in-one settings demonstrate the effectiveness of our model.

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

## APPENDIX

This appendix provides more experimental results, ablation studies, and visual comparisons.

## A    MORE EXPERIMENTAL RESULTS

In this section, we first provide experimental results on LOL-V2 (Yang et al., 2021b) for low-light image enhancement. The numerical results are presented in Table 11. Our method significantly outperforms the Transformer-based algorithm Retinexformer (Cai et al., 2023) by 0.43 dB PSNR. The visual results are illustrated in Figure 10. Our model recovers more edges from the input image. These results suggest the strong potential of our method for low-light image enhancement.

Table 11: Numerical comparisons on the LOL-V2-synthetic dataset (Yang et al., 2021b) for low-light image enhancement.

| Methods | PSNR | SSIM |
|---|---|---|
| RUAS (Liu et al., 2021a) | 16.55 | 0.652 |
| FIDE (Xu et al., 2020) | 15.20 | 0.612 |
| DRBN (Yang et al., 2021a) | 23.22 | 0.927 |
| KinD (Zhang et al., 2019b) | 13.29 | 0.578 |
| Restormer (Zamir et al., 2022a) | 21.41 | 0.830 |
| MIRNet (Zamir et al., 2022b) | 21.94 | 0.876 |
| SNR-Net (Xu et al., 2022) | 24.14 | 0.928 |
| Retinexformer (Cai et al., 2023) | 25.67 | 0.930 |
| **Ours** | **26.10** | **0.944** |

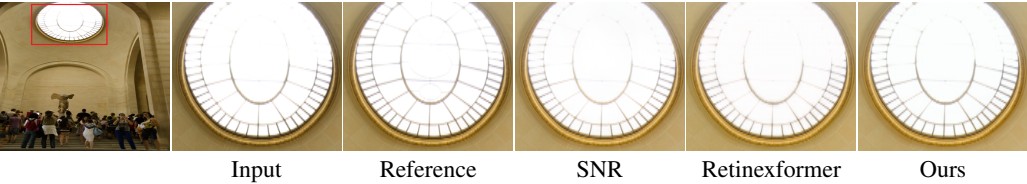

| Input | Reference | SNR | Retinexformer | Ours |

Figure 10: Visual results on LOL-V2-Synthetic (Yang et al., 2021b).

Table 12: The numerical comparisons on five image restoration tasks under the all-in-one setting: dehazing (SOTS (Li et al., 2018)), deraining (Rain100L (Yang et al., 2017)), denoising (BSD68 (Martin et al., 2001)), deblurring (GoPro (Nah et al., 2017)), and low-light image enhancement (LOL-V1 (Wei et al., 2018)). The results are reported in the form of PSNR/SSIM.

| Method | Dehazing | Deraining | Denoising | Deblurring | Low-Light | Average |
|---|---|---|---|---|---|---|
| NAFNet (Chen et al., 2022) | 25.23/0.939 | 35.56/0.967 | 31.02/0.883 | 26.53/0.808 | 20.49/0.809 | 27.76/0.881 |
| MPRNet (Zamir et al., 2021) | 24.27/0.937 | **38.16**/0.981 | 31.35/0.889 | 26.87/0.823 | 20.84/0.824 | 28.27/0.890 |
| MIRNetV2 (Zamir et al., 2022b) | 24.03/0.927 | 33.89/0.954 | 30.97/0.881 | 26.30/0.799 | 21.52/0.815 | 27.34/0.875 |
| SwinIR (Liang et al., 2021) | 21.50/0.891 | 30.78/0.923 | 30.59/0.868 | 24.52/0.773 | 17.81/0.723 | 25.04/0.835 |
| Restormer (Zamir et al., 2022a) | 24.09/0.927 | 34.81/0.962 | 31.49/0.884 | 27.22/0.829 | 20.41/0.806 | 27.60/0.881 |
| DL (Fan et al., 2019) | 20.54/0.826 | 21.96/0.762 | 23.09/0.745 | 19.86/0.672 | 19.83/0.712 | 21.05/0.743 |
| Transweather (Valanarasu et al., 2022) | 21.32/0.885 | 29.43/0.905 | 29.00/0.841 | 25.12/0.757 | 21.21/0.792 | 25.22/0.836 |
| TAPE (Liu et al., 2022) | 22.16/0.861 | 29.67/0.904 | 30.18/0.855 | 24.47/0.763 | 18.97/0.621 | 25.09/0.801 |
| AirNet (Li et al., 2022) | 21.04/0.884 | 32.98/0.951 | 30.91/0.882 | 24.35/0.781 | 18.18/0.735 | 25.49/0.846 |
| IDR (Zhang et al., 2023) | 25.24/0.943 | 35.63/0.965 | **31.60**/0.887 | 27.87/0.846 | 21.34/0.826 | 28.34/0.893 |
| **Ours** | **30.29/0.978** | 38.08/**0.982** | 31.37/**0.891** | **28.31/0.860** | **22.89/0.855** | **30.19/0.913** |

Moreover, we report experimental results under all-in-one image restoration, *i.e.,* five-task setting. The quantitative results are presented in Table 12. As seen, our method achieves a PSNR score of

30.19 when averaging across all tasks, which is 1.85 dB higher than that of IDR (Zhang et al., 2023). In particular, for the dehazing problem, our model significantly outperforms the second-best algorithm (Zhang et al., 2023) by 5.05 dB PSNR. Despite not incorporating a complex dynamic mechanism for identifying degradation types, aside from SA, our method consistently delivers promising results across various all-in-one tasks, thanks to its robust representational capability.

Table 13: Ablation studies of the deployment strategy for different kinds of attention.

| Scale 0 | Scale 1 | Scale 2 | PSNR |
|---------|---------|---------|------|
| Spatial | Spatial | Spatial | 31.76 |
| Channel | Spatial | Spatial | 31.82 |
| Channel | Channel | Spatial | 31.62 |
| Channel | Channel | Channel | 31.55 |
| Normal Attention (Zamir et al., 2022a) | | | 31.50 |

Table 14: More ablation studies for DFFN.

| Methods | PSNR |
|---------|------|
| Ours | 31.82 |
| only frequency branch in spatial-spectral interactions | 31.76 |
| w/o attention weight for spatial-spectral fusion | 31.73 |

## B   MORE ABLATION STUDIES

**Deployment strategy for attention.** We apply channel-wise modulation block in the first scale while using spatial-wise block in other scales, as spatial-wise SA is more expensive than channel version when modeling large-scale features. In our case, the first scale includes the highest-resolution features. Table 13 shows that our strategy achieves the best performance. Moreover, we experiment by using only regular channel attention Zamir et al. (2022a) in all scales, achieving a 0.32 dB lower performance than our full model. These results validate the efficacy of our design.

**DFFN.** We conduct more ablation studies for DFFN by removing or substituting certain operators. Table 14 shows that removing the spatial branch in inter-domain fusion achieves 31.76 dB PSNR, suggesting the significance of dual-domain feature fusion. Removing the attention weights leads to 31.73 dB PSNR, which is even lower than the result of using a single branch, *e.g.,* frequency branch (31.76 dB), demonstrating the importance of coordinating the fusion process.

**Modulation design.** In this part, we perform ablation studies for the modulation design. We use the plain depth-wise convolutions with the same kernel size to supplant the filter operation, achieving 31.69 dB PSNR, which is 0.13 dB lower than our design.

**Parameter-sharing mechanism.** In our model, we share the parameters across CMB. We carry out experiments to apply the parameter-sharing strategy in deeper scales, achieving lower performance than our design (see Table 15). We also attempt to further share the parameters among DFFN in the first scale, obtaining only 30.53 dB PSNR. Therefore, we only apply the mechanism in CMB for better performance.

**Position of downsampling.** In CMB, we apply downsampling after the convolutions, which can fully learn the spatial connectivity, as the channel-wise SA layer cannot model the real spatial pixel interactions. We experiment by moving downsampling before convolutions, saving 1.89 GFLOPs while achieving 0.11 dB lower PSNR. Finally, we choose to place downsampling after convolutions in the CMB of our model.

Table 15: Abltion studies for the parameter-sharing mechanism. Scale 0,1,2 means sharing parameters within each scale of all scales.

| Method | PSNR |
|---|---|
| Scale 0 | 31.82 |
| Scale 0,1 | 31.82 |
| Scale 0,1,2 | 31.82 |

## C MORE VISUAL RESULTS

Visual comparisons on more datasets are illustrated in Figure 11 and Figure 12.

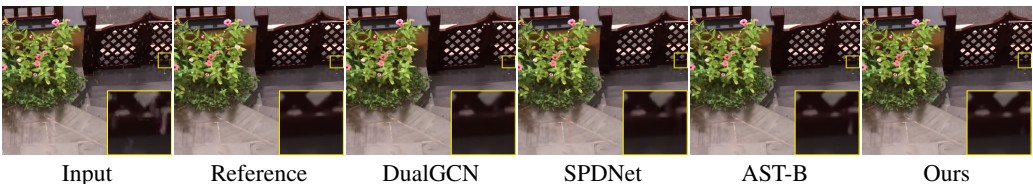

Input   Reference   DualGCN   SPDNet   AST-B   Ours

Figure 11: Image deraining comparisons on the SPAD (Wang et al., 2019) dataset.

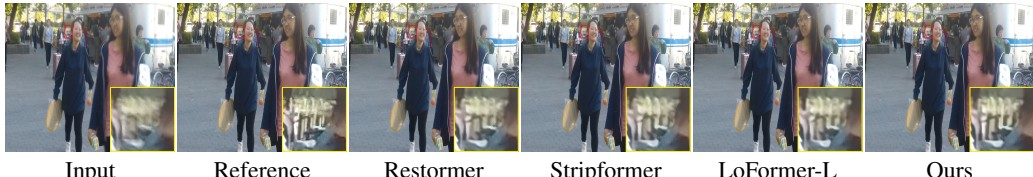

Input   Reference   Restormer   Stripformer   LoFormer-L   Ours

Figure 12: Motion deblurring comparisons on the HIDE (Shen et al., 2019) dataset.

