# OpenReview forum: "Modumer: Modulating Transformer for Image Restoration"
_ICLR.cc/2025/Conference — ICLR 2025 Conference Withdrawn Submission_

### Official Review · Reviewer_bAcY · 2024-10-29

**Soundness:** 3
**Presentation:** 2
**Contribution:** 2
**Rating:** 3
**Confidence:** 5

**Summary:**

This paper develops Modumer for effective and efficient image restoration by revisiting the Transformer block and Modulation design. It presents a novel attention block that consecutively modulates the self-attention results from downsampled features, providing efficient omni-kernel modulation and high-dimensional representational capability. Additionally, a bio-inspired parameter-sharing mechanism is introduced in the attention layers to improve both efficiency and performance.

**Strengths:**

1. This paper introduces Modumer, proposing a novel approach by revisiting the Transformer block and modulation design, significantly enhancing the self-attention mechanism and innovatively incorporating a bio-inspired parameter-sharing technique. This new perspective makes an important contribution to the field of image restoration.
2. The paper is well-structured, with key ideas expressed clearly, effectively elucidating the proposed methods.

**Weaknesses:**

1. The paper does not provide detailed ablation studies, such as those for CMB and SMB. Additionally, while it mentions that the bio-inspired parameter-sharing mechanism and novel attention block may positively impact performance, it lacks strong analysis or evidence to justify the design choices made.

2. Although the paper claims to achieve state-of-the-art performance, it lacks detailed comparisons with a wider range of updated all-in-one methods, particularly those published in the last year. For instance, comparisons with recent methods like AutoDIR (AutoDIR: Automatic All-in-One Image Restoration with Latent Diffusion) and InstructIR (InstructIR: High-Quality Image Restoration Following Human Instructions) are missing. Furthermore, the paper does not provide a more comprehensive visual comparison.

3. The validation is confined to public datasets and does not include testing on real benchmark datasets, which is insufficient to demonstrate the effectiveness and generalizability of the proposed method. The experiments primarily focus on standard image restoration tasks，exploring the applicability of Modumer in real-world scenarios or other fields would be beneficial.

**Questions:**

1. Could you provide a detailed explanation of the effectiveness of the attention mechanism in Modumer and its specific advantages? Can you offer any quantitative or qualitative analysis to support this?

2. What is the rationale behind using the DFFN component in your model? Why did you choose DFFN instead of directly using the feedforward layer (FFL) from the Transformer? Please elaborate on your considerations and the pros and cons of this choice.

3. The paper mentions two model versions, Modumer-S and Modumer-B. In Table 10, which model is represented by the "Ours" method? Please clarify this to help readers better understand the source of the results.

---

### Official Review · Reviewer_kNuU · 2024-10-30

**Soundness:** 2
**Presentation:** 2
**Contribution:** 2
**Rating:** 5
**Confidence:** 5

**Summary:**

This paper revisits the Transformer block and modulation design, proposing a combination block. Additionally, a parameter-sharing mechanism and a dual-domain feed-forward network are introduced. The experimental results show good improvements provided by the authors.

**Strengths:**

1. The paper is well-written and easy to read.
2. Performance gains have achieved on the datasets provided by the authors.

**Weaknesses:**

The paper appears to primarily combine existing techniques without introducing substantial new insights. The following points outline key concerns:
1.	The proposed attention block is largely a combination of the Transformer block and modulation design, with limited novelty.
2.	The parameter-sharing mechanism is meaningless. Table 8 also demonstrates the marginal improvements.
3.	The differences between the proposed dual-domain feed-forward network and the FFN in FFTformer [1] are not clearly articulated.

Additional concerns include:
4.	The challenges addressed by the proposed method are not clearly defined. Is the goal solely to improve PSNR? A more thorough discussion of the objectives and the challenges this approach aims to solve would strengthen the paper.
5.	The baseline comparisons could be more comprehensive. It appears that only favorable results are presented, potentially overlooking stronger competitors. For example, Table 4 omits recent state-of-the-art methods from CVPR 2024, such as AdaRevD (34.64 dB) [2] and SegFFTformer (34.48 dB) [3], which outperform the proposed method on GoPro.

[1] Efficient Frequency Domain-based Transformers for High-Quality Image Deblurring
[2] AdaRevD: Adaptive Patch Exiting Reversible Decoder Pushes the Limit of Image Deblurring
[3] Real-World Efficient Blind Motion Deblurring via Blur Pixel Discretization.

**Questions:**

1.	An inconsistency is observed in Figure 2, where the x-axis differs from other figures, impacting clarity.
2.	Please clarify how Eq. 8 specifically contributes to the image restoration process.
3.	Baselines like FSNet and ConvIR have conducted experiments on various tasks, including defocus deblurring (DPDD) and image dehazing (ITS, OTS, DenseHaze). Could the authors provide the corresponding results for these tasks?
4.	Results on RealBlur-R/J datasets can be provided and would provide a fuller comparison.
5.     Could the authors clarify why the performance of Ours-S surpasses that of FSNet and ConvIR (Base model) in Table 7?

---

### Official Review · Reviewer_X9oQ · 2024-11-02

**Soundness:** 2
**Presentation:** 2
**Contribution:** 2
**Rating:** 3
**Confidence:** 5

**Summary:**

The authors introduce a Transformer-based image restoration network that leverages an efficient block design and parameter sharing to minimize overall computational complexity. The core concept revolves around modulated Transformers, including attention blocks enhanced with additional convolutions to scale the output features. This approach achieves competitive or superior performance compared to prior methods on various restoration tasks, including image deraining, motion deblurring, dehazing, and desnowing.

**Strengths:**

1. The proposed architecture demonstrates good performance across various image restoration tasks, including raindrop removal and motion deblurring.

2. The high-level design incorporates notable concepts, such as feature downscaling for efficient processing, parameter sharing, channel-wise operations in the upper UNet layers, and spatially focused operations in the deeper layers.

**Weaknesses:**

1. The main motivation of this paper is unclear; the authors focus on incremental architecture engineering without clearly explaining the reasoning behind these modifications.

2. The proposed architecture shows limited originality and appears more as a combination of components from prior work, such as the DFFN module introduced by FFTTransformer [ref1] for image deblurring, yet the authors present it as their own contribution. Furthermore, the SMB block closely resembles the RGT [ref2] structure. Could be please clarify how your approach differs from these specific prior works?

3. The motivation for parameter sharing is unclear, with no compelling rationale provided for why this approach should be effective. Additionally, the ablation studies lack a comparison of results between networks with and without parameter sharing, leaving its impact unexplored.

4. The authors claim that the proposed architecture design is more efficient than previous work, however this is not supported by the presented results, e.g. Tab4 and Tab7 shows that the networks uses more FLOPS than previous work.

[ref1] Efficient Frequency Domain-based Transformer for High-Quality Image Deblurring, CVPR 2023

[ref2] Chen et al., Recursive Generalization Transformer for Image Super-Resolution, ICLR 2024

**Questions:**

1. Why are the features in the channel attention block spatially downsampled, given that the attention mechanism operates in the channel domain?

2. Why is the attention block described as the "shared" memory of the network, when the feed-forward block actually contains a significantly larger number of parameters than the attention blocks?

3. Why are the GFLOPs relatively high compared to other works, despite the proposed network performing numerous operations on lower-resolution feature maps, which should theoretically improve overall efficiency?

4. The use of extreme downscaling factors is unusual, as the features in the U-Net architecture are already reduced in size. Why not consider using efficient convolutions in encoder/decoder and self-attention operations only in the bottleneck instead?

---

### Official Review · Reviewer_qrd9 · 2024-11-03

**Soundness:** 2
**Presentation:** 2
**Contribution:** 2
**Rating:** 3
**Confidence:** 5

**Summary:**

A modulated framework for image restoration.The performance is good but the architecture is not so novel and the experimental settings are not common.

**Strengths:**

1. Sufficient experiments done in the manuscript.
2. The results on several benchmarks are good.

**Weaknesses:**

1. The proposed modules lack novelty. The whole framework is based on modulation and Transformer. Both of them are not newly proposed in this manuscript, and attention blocks in the manuscript are combination of two staffs.
2. Some experimental settings are confusing, especially on the general image restoration. Common tasks for image restoration, like denoising and SR, are missing.
3. Experimental comparison on some tasks may be unfair. Two size models of proposed method are used for evaluation, and they have huge computational cost differences.

**Questions:**

1. The proposed architecture is not novel enough. The authors should provide insights and reasons of using modulation and dual-domain FFN. Please clarify motivations and expected benefits of combining modulation with dual-domain FFN in this particular way for image restoration tasks.
2. The proposed method is divided into two versions, small and base. How do the authors design two versions? Why do the authors choose the base version for the deblurring task?
3. The models is used to deal with several common used tasks and the authors title this manuscript as for image restoration. However, for the general restoration, the most used two tasks: denoising and SR are neglected. It is doubtful why the authors choose to avoid them in the experiments.
4. For the all-in-one settings, it would be better if the authors can focus on those all-in-one methods rather than some general methods, such as MPRNet. As there are many all-in-one methods now, the authors should refer to some new results like tables in [1] and [2].
5. Table 9 is not well written. Some typos like denoising should be avoided. The details of those datasets are not illustrated as those data may be splitted manually. Also, three tasks are used in the main paper, but five tasks in the appendix, so the authors should demonstrate the two kinds of settings rather than in single table.
6. Some results also show the Params and FLOPs in the manuscript. What about the all-in-one settings? For the all-in-one settings, the computational cost is more sensitive.
7. The proposed method is not designed especially for all-in-one. Why do the authors not only focus on general but also all-in-one?

Final scores can be modified upon replies.

[1] Perceive-IR: Learning to Perceive Degradation Better for All-in-One Image Restoration. X Zhang, J Ma, G Wang, Q Zhang, H Zhang, L Zhang
[2] AdaIR: Adaptive All-in-One Image Restoration via Frequency Mining and Modulation. Y Cui, SW Zamir, S Khan, A Knoll, M Shah, FS Khan

---

### Note · Authors · 2024-11-12

I have read and agree with the venue's withdrawal policy on behalf of myself and my co-authors.